**www.cambridge.org/qrd**

## Research Article

Autoencoders; collective variables; free-energy calculations; slow modes; VAMPnets

**Author for correspondence:**
*Christophe Chipot,
E-mail: chipot@illinois.edu

# Chasing collective variables using temporal data-driven strategies

Haochuan Chen[1] and Christophe Chipot[1,2,3]*

[1]Laboratoire International Associé Centre National de la Recherche Scientifique et University of Illinois at Urbana-Champaign, Unité Mixte de Recherche n°7019, Université de Lorraine, 54506 Vandœuvre-lès-Nancy, France; [2]Theoretical and Computational Biophysics Group, Beckman Institute, and Department of Physics, University of Illinois at Urbana-Champaign, Urbana, IL 61801, USA and [3]Department of Biochemistry and Molecular Biology, University of Chicago, Chicago, IL 60637, USA

## Abstract

The convergence of free-energy calculations based on importance sampling depends heavily on the choice of collective variables (CVs), which in principle, should include the slow degrees of freedom of the biological processes to be investigated. Autoencoders (AEs), as emerging data-driven dimension reduction tools, have been utilised for discovering CVs. AEs, however, are often treated as black boxes, and what AEs actually encode during training, and whether the latent variables from encoders are suitable as CVs for further free-energy calculations remains unknown. In this contribution, we review AEs and their time-series-based variants, including time-lagged AEs (TAEs) and modified TAEs, as well as the closely related model variational approach for Markov processes networks (VAMPnets). We then show through numerical examples that AEs learn the high-variance modes instead of the slow modes. In stark contrast, time series-based models are able to capture the slow modes. Moreover, both modified TAEs with extensions from slow feature analysis and the state-free reversible VAMPnets (SRVs) can yield orthogonal multidimensional CVs. As an illustration, we employ SRVs to discover the CVs of the isomerizations of $N$-acetyl-$N'$-methylalanylamide and trialanine by iterative learning with trajectories from biased simulations. Last, through numerical experiments with anisotropic diffusion, we investigate the potential relationship of time-series-based models and committor probabilities.

## Introduction

Obtaining the correct free-energy landscapes, and possibly the dynamics underlying biological processes remains a challenging task for computer simulations, since not only accurate force fields are required, but also ergodic sampling of the configurational space, which is often hindered by high free-energy barriers. To accelerate sampling, a variety of enhanced-sampling strategies (Abrams and Bussi, 2013; Chen and Chipot, 2022; Hénin *et al.,* 2022) have been developed to apply biasing forces or potentials onto a surrogate of the reaction coordinate (RC), which chiefly consists of collective variables (CVs) able to achieve time scale separation or capture the essential slow degrees of freedom (DOFs) of the underlying mechanics of the process of interest. Discovery of suitable CVs has been traditionally guided by chemical and physical intuitions. For example, a Euclidian distance and a set of Euler and polar angles have proven appropriate CVs to describe the relative position and orientation of the binding partners in protein-ligand complexes (Gumbart *et al.,* 2013; Fu *et al.,* 2017). This approach, however, requires in-depth knowledge of the biological process at hand, rationalising the absence of standardised protocols for the design of suitable CVs for a broader gamut of biological processes. In order to address this difficulty, a variety of data-driven and machine-learning approaches have been explored, including, albeit not limited to, principal component analysis (PCA) (Pearson, 1901; Altis *et al.,* 2007; Maisuradze *et al.,* 2009), time-structure based independent component analysis (tICA) (Molgedey and Schuster, 1994; Naritomi and Fuchigami, 2011), and spectral gap optimization of order parameters (SGOOP) (Tiwary and Berne, 2016). Inspired by the development of artificial intelligence, data-driven approaches have been extended and melded with deep neural networks (DNNs), most notably autoencoders (AEs) (Hinton and Salakhutdinov, 2006), for example, molecular enhanced sampling with AEs (MESA) (Chen and Ferguson, 2018), AEs in machine-learning collective variable (MLCV) (Chen *et al.,* 2022a), free-energy biasing and iterative learning with AEs (FEBILAE) (Belkacemi *et al.,* 2022), and extended AEs (Frassek *et al.,* 2021), to name but a few. These classical AE-based schemes are powerful for constructing a low-dimensional latent space from atomic coordinates, or known CVs, which maximises the explained variances. The temporal information of the simulation trajectories is, however, ignored in classical AEs. A number of strategies have been put forth to take time explicitly into account in NN-based models, which includes, but is not limited to variational approach for Markov processes networks (VAMPnets) (Mardt *et al.,* 2018), time-lagged AEs (TAEs) (Hernández

*et al.,* 2018; Wehmeyer and Noé, 2018), modified TAEs (Chen *et al.,* 2019*a*), past–future information bottleneck (PIB) (Wang *et al.,* 2019) and log-probability estimation via invertible NN for enhanced sampling (LINES) (Odstrcil *et al.,* 2022). As we will show in our numerical experiments, the variables that can maximise the explained variances do not always necessarily coincide with the important DOFs of the process of interest. This observation raises a question – are classical AEs always able to identify the relevant CVs? If not, could time series-based models solve the problem? In this contribution, we review the use of classical AEs, TAEs, modified TAEs, VAMPnets as well as their variants for CV discovery in an attempt to address these questions. In addition, we illustrate in a series of numerical experiments the strengths and limitations of CV-discovery methods based on classical AEs for the identification of relevant DOFs in molecular processes. Next, we show how to overcome these limitations by turning to time-series-based techniques, such as TAEs, modified TAEs and state-free reversible VAMPnets (SRVs) (Chen *et al.,* 2019*b*). Finally, we use trajectories from biased simulations of *N*-acetyl-*N′*-methyl-alanylamide (NANMA, also known as alanine dipeptide) and a terminally blocked trialanine as training data as examples of CV discovery by iterative learning using SRVs. We also outline the potential relationship between the committor probability (Geissler *et al.,* 1999; Berezhkovskii and Szabo, 2019) and the latent variable learned from time-series-based techniques.

## Methods

In this section, we review the AEs that have been employed for the purpose of CV discovery, with an emphasis on the comparison of classical AEs, TAEs, modified TAEs, as well as VAMPnets and SRVs, which are closely related to TAEs (Wehmeyer and Noé, 2018). We further elaborate on the implications of turning to TAEs and possibly to SRVs, as opposed to mere AEs, and why for specific applications, it is crucial to introduce a temporal component to the problem at hand. Next, we detail our iterative strategy leaning on biased trajectories to train the AEs, and rationalise why biasing is of paramount importance.

### Classical AEs

Consider a simulation trajectory of a molecular object of $n$ atoms with coordinates $\mathbf{x}(t) \equiv \mathbf{x}_1(t), \cdots, \mathbf{x}_n(t)$, and a transformation $\mathbf{s}(t)$ of $m$ component, namely $\mathbf{s}(t) \equiv s_1(\mathbf{x}(t)), \cdots, s_m(\mathbf{x}(t))$. These transformations can be choice functions mapping to a single component of a coordinate such as $s_1(t) \equiv x_{1x}(t)$, or some nonlinear functions mapping atomic coordinates to internal variables like dihedral angles, as long as the derivatives $\nabla_{\mathbf{x}}\mathbf{s}$ are well defined. An AE (Hinton and Salakhutdinov, 2006; Goodfellow *et al.,* 2016), as illustrated in *Fig. 1a*, can be regarded as a multilayer NN, which consists of an encoder part and a decoder part. In previous work employing AEs in CV discovery (Chen and Ferguson, 2018; Chen *et al.,* 2018; Belkacemi *et al.,* 2022; Chen and Chipot, 2022), the inputs of the encoder are $\mathbf{s}(t)$, and the encoder transforms them into a vector of latent variables $\boldsymbol{\xi} \equiv \xi_1, \cdots, \xi_d$, which are then decoded into $\hat{\mathbf{s}}(t)$ by the decoder. In general, for the purpose of dimensionality reduction, $d$ is much smaller than $m$. A typical loss function – that is, a real-valued function for measuring the performance of a NN, $\mathscr{L}$, is equal for the classical AEs to the mean-squared-error (MSE), namely,

$$\mathscr{L} = \frac{1}{N}\sum_{j=0}^{N}\|\mathbf{s}(j\Delta t) - \hat{\mathbf{s}}(j\Delta t)\|^2, \tag{1}$$

where $\Delta t$ is the time interval for discretizing and saving the trajectory, and $N$ is the number of frames of the trajectory. It ought to be noted that since PCA can be interpreted as finding directions for projecting the data sets that either maximise the variance or minimise the reconstruction error, an AE that uses linear activation functions for all layers and MSE loss can strictly speaking be approximated to PCA. Nonlinear AEs resting on nonlinear activation functions can be regarded as a generalisation of PCA. Methods based on classical AEs have been applied to explore the conformational changes of chignolin (Belkacemi *et al.,* 2022) and Trp–cage (Chen and Ferguson, 2018), to analyse the dynamic allostery triggered by ligand binding or mutagenesis (Tsuchiya *et al.,* 2019), christo decode the conformational heterogeneities underlying the cytochrome P450 protein (Bandyopadhyay and Mondal, 2021), and to cluster the folded states of the $\beta\beta\alpha$–protein (Ghorbani *et al.,* 2021).

### Time-lagged AEs

The slow modes of molecular dynamics (MD) trajectories could be identified by exploiting their Markovianity and modelling the dynamics by means of a master equation (Mezić, 2005; Mitsutake *et al.,* 2011; Noé and Nüske, 2013; Husic and Pande, 2018). From this perspective, various efforts have underscored that the slow modes can be expressed as the eigenfunctions of the Markov transition-rate matrix – or operator, and finding these eigenfunctions is tantamount to maximising the following functional in a variational principle sense (Takano and Miyashita, 1995; Noé and Nüske, 2013):

$$R[f_\theta] = \frac{\langle f_\theta(\mathbf{x}(t)) f_\theta(\mathbf{x}(t+\tau)) \rangle}{\langle f_\theta(\mathbf{x}(t)) f_\theta(\mathbf{x}(t)) \rangle}, \tag{2}$$

where $f_\theta$ are the eigenfunctions, and $\tau$ is the time lag. From a different point of view, Eq. (2) can be regarded as a normalised autocorrelation function, and the discovery of slow modes can be done by maximising it (Schwantes and Pande, 2013). McGibbon *et al.* (2017) showed that the second leading eigenfunction of the transfer operator could be used as an appropriate CV – which they term a natural reaction coordinate since it is the most slowly decorrelating mode and maximally predictive of future evolution (Sultan and Pande, 2017). Akin to the relationship between PCA and AEs, maximisation of the correlation function in Eq. (2) can be approximated as the minimization of reconstruction loss between $f_\theta(\mathbf{x}(t))$ and $f_\theta(\mathbf{x}(t+\tau))$ by means of a regression approach, which leads to a TAE (Wehmeyer and Noé, 2018). In other words, assuming the time lag $\tau$ can be discretized as $\alpha\Delta t$, and $\mathbf{s}(t)$ is whitened (Kessy *et al.,* 2018) by zero-phase components analysis (Bell and Sejnowski, 1997), the loss function in TAEs is:

$$\mathscr{L} = \frac{1}{N-\alpha}\sum_{j=0}^{N-\alpha}\|\mathbf{s}(j\Delta t + \alpha\Delta t) - \hat{\mathbf{s}}(j\Delta t)\|^2. \tag{3}$$

TAEs can be considered as NN-based extensions (see *Fig. 1a*) to the dynamical mode decomposition (DMD) (Schmid, 2010) and time-lagged canonical correlation analysis (TCCA) (Hotelling, 1936). TAEs have been successfully applied to study the conformational changes of villin (Wehmeyer and Noé, 2018), and its extension by variational AE (Hernández *et al.,* 2018) has been employed to train transferable CVs between one protein and its mutants (Sultan *et al.,* 2018).

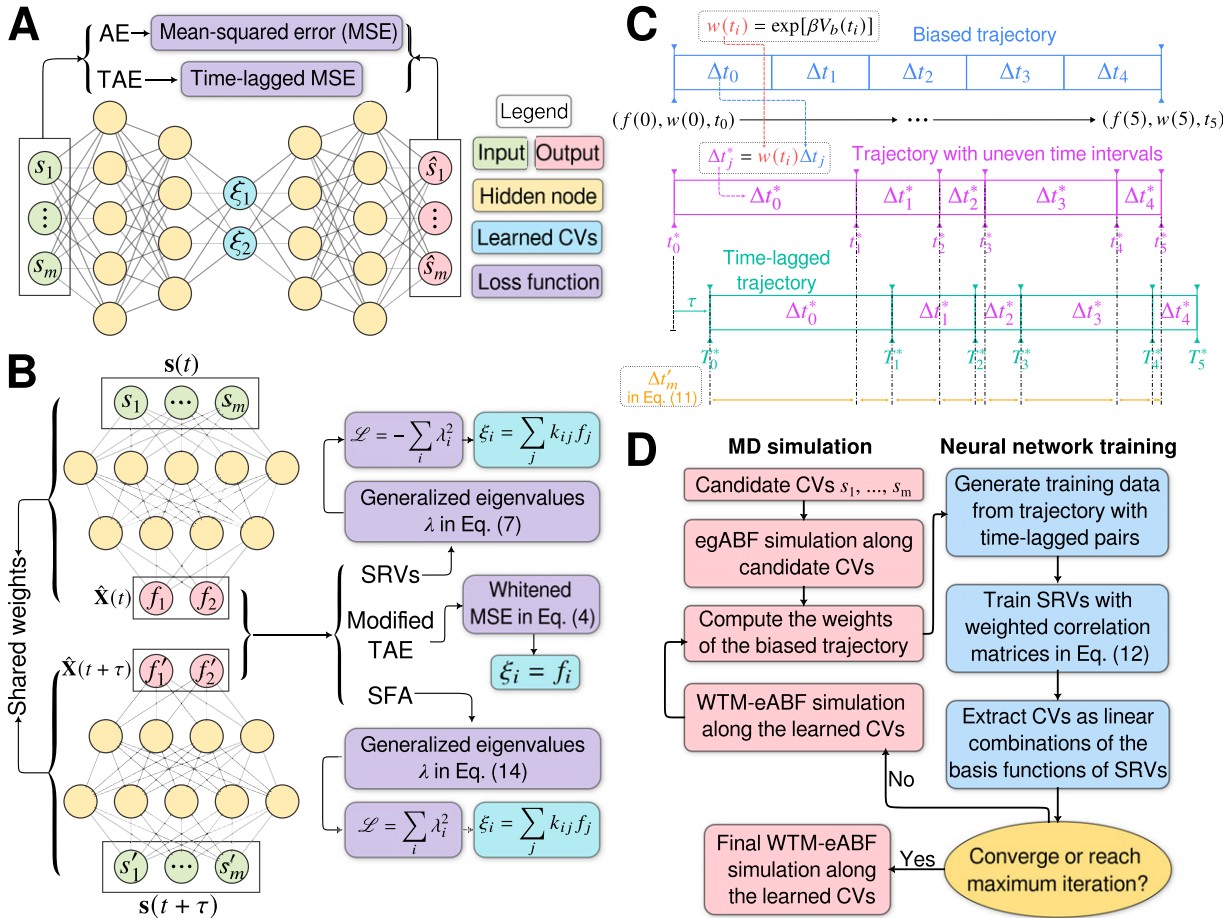

**Fig. 1.** (*a*) Schematic representation of a neural network used in an autoencoder (AE), or in a time-lagged autoencoder (TAE). (*b*) Schematic representation of a Siamese neural network used in modified TAEs, in state-free reversible VAMPnets (SRVs), and in a slow feature analysis (SFA). (*c*) Calculation of the reweighting factor $\Delta t'_m$ in Eq. (11). (*d*) Workflow employed in this work of data-driven collective-variable (CV) discovery from biased molecular dynamics (MD) simulations.

## Modified TAEs

Chen *et al.* (2019*a*) revealed that TAEs with nonlinear activation functions tend to learn a mixture of slow modes, as well as the maximum-variance mode, and in some cases the slowest mode could be missing, leading to suboptimal CVs. To overcome this limitation, they proposed a modified TAE in the same work, which uses similar Siamese NNs (see *Fig. 1b*) as VAMPnets (Mardt *et al.*, 2018), alongside its state-free reversible variant (Chen *et al.*, 2019*b*). The loss function of modified TAEs is

$$\mathscr{L} = \frac{\sum_{j=0}^{N-\alpha} \|\hat{\boldsymbol{\xi}}(j\Delta t + \alpha\Delta t) - \hat{\boldsymbol{\xi}}(j\Delta t)\|^2}{(N-\alpha)\sigma^2(\hat{\boldsymbol{\xi}}(j\Delta t))}, \tag{4}$$

where the $\sigma^2\left(\hat{\boldsymbol{\xi}}(j\Delta t)\right)$ is the variance of the encoder output. In addition, Chen *et al.* (2019*a*) also showed that for those cases where $m = 1$, the modified TAEs are equivalent to the SRVs (Chen *et al.*, 2019*b*), albeit the latter can guarantee the orthogonality of the latent variables by employing a variational approach in Eq. (2) with an eigendecomposition, as will be discussed in the following section.

## VAMPnets and its state-free reversible derivative

VAMPnets (Mardt *et al.*, 2018) employ the same Siamese NN (Bromley *et al.*, 1993; Chicco, 2021) (see *Fig. 1b*) as discussed

previously, but instead of turning to the regression approach of Eq. (3), they solve the variational problem in Eq. (2) by employing TCCA on the encoded features. Assuming that the encoder can be expressed as a zero-mean multivariate basis function, $\mathbf{f}(\mathbf{s}(t))$, the loss function of VAMPnets is,

$$\mathscr{L} = \left\|\mathbf{C}(t,t)^{-\frac{1}{2}}\mathbf{C}(t,t+\tau)\mathbf{C}(t+\tau,t+\tau)^{-\frac{1}{2}}\right\|_F^2, \tag{5}$$

where $\|\cdot\|_F$ denotes the Frobenius norm of a matrix, and the correlation matrices $\mathbf{C}(t,t)$, $\mathbf{C}(t,t+\tau)$ and $\mathbf{C}(t+\tau,t+\tau)$ (Wu and Noé, 2020) are defined, respectively, as:

$$\begin{cases} \mathbf{C}(t,t) & = \frac{1}{N-\alpha}\hat{\mathbf{X}}(t)\hat{\mathbf{X}}(t)^{\mathrm{T}} \\ \mathbf{C}(t,t+\tau) & = \frac{1}{N-\alpha}\hat{\mathbf{X}}(t)\hat{\mathbf{X}}(t+\tau)^{\mathrm{T}} \\ \mathbf{C}(t+\tau,t+\tau) & = \frac{1}{N-\alpha}\hat{\mathbf{X}}(t+\tau)\hat{\mathbf{X}}(t+\tau)^{\mathrm{T}}, \end{cases} \tag{6}$$

where the elements in $\hat{\mathbf{X}}(t)$ and $\hat{\mathbf{X}}(t+\tau)$ are $\hat{X}_{ij}(t) = f_i(\mathbf{s}(j\Delta t))$ and $\hat{X}_{ij}(t+\tau) = f_i(\mathbf{s}(j\Delta t + \alpha\Delta t))$, respectively. If the dynamics are reversible, which is the case in equilibrium MD simulations, the loss function relates to the eigenvalues of the Koopman operator of the Markov process by $\mathscr{L} = -\sum_{i=1}^{m}\lambda_i^2$, where $\lambda_i$ is the eigenvalues

solved from the following generalised eigenvalue problem (Schwantes and Pande, 2013):

$$\mathbf{C}(t, t+\tau)\mathbf{V} = \mathbf{C}(t,t)\mathbf{V}\mathbf{\Lambda}. \tag{7}$$

Here, $\mathbf{\Lambda}$ is $\text{diag}(\lambda_1, \cdots, \lambda_m)$, and $\mathbf{V}$ is a matrix containing the eigenvectors. The optimization of the loss function can be regarded as maximising the total kinetic variance (Noé and Clementi, 2015). After optimization, the learned CVs can be obtained by linear combinations of the components of $\mathbf{f}$, with the weights from the eigenvectors, namely $\xi_i = \sum_j k_{ij} f_j$, where $k_{ij}$ is the -$j$th component of the -$i$th column vector of $\mathbf{V}$. The number of learned CVs should not be greater than the number of components of $\mathbf{f}$. The use of Eq. (7) with basis function $\mathbf{f}$ can be viewed as an extension of tICA (Naritomi and Fuchigami, 2011), with a kernel function (Schwantes and Pande, 2015), while the kernel functions are approximated by the NNs. This loss function is also employed by both SRVs (Chen *et al.,* 2019*b*) and deep-TICA (Bonati *et al.,* 2021). It ought to be noted that while the principle of Eqs. (6) and (7) is generally considered to have been proposed as early as 1994 by Molgedey and Schuster (1994), a similar methodology has been devised independently in many other fields under different names, such as relaxation mode analysis (Takano and Miyashita, 1995; Mitsutake *et al.,* 2011), second-order independent component analysis (Belouchrani *et al.,* 1997), and temporal decorrelation source separation (Ziehe and Müller, 1998). VAMPnets have been utilised to find metastable states in the folding of the N-terminal domain of ribosomal protein L9 (NTL9) (Mardt *et al.,* 2018), to determine a kinetic ensemble of the 42-amino-acid amyloid-$\beta$ peptide (Löhr *et al.,* 2021). In addition, its state-free reversible variant has been employed for identifying the metastable states of DNA hybridization/dehybridization (Jones *et al.,* 2021).

### *Iterative learning with a biased trajectory*

A common shortcoming of all data-driven dimensionality reduction methods is that the learned models may perform poorly on unseen data. To be specific, if the states of interest are not part of the training set, the learned CVs could lose the information about them. To address this difficulty, one can turn to an iterative learning strategy, that is, one can run the following iterations until some convergence criteria are met, namely, (a) feeding the model with an initial trajectory, (b) biasing the next round of the simulation with the learned CVs, and (c) using the biased trajectory to train the model again. This strategy has been employed in MESA (Chen and Ferguson, 2018), FEBILAE (Belkacemi *et al.,* 2022), deep-TICA (Bonati *et al.,* 2021) and PIB (Wang *et al.,* 2019). Specifically, in this contribution, we start by training the SRVs with an initial trajectory from extended generalised adaptive biasing force (egABF) (Zhao *et al.,* 2017) simulations along candidate CVs chosen coarsely by physicochemical intuition – that is, educated guesses, and employ well-tempered meta-extended ABF (WTM-eABF) (Fu *et al.,* 2019) simulations along the learned CVs in successive iterations, as depicted in *Fig. 1d*. A key issue of iterative learning is the reweighting of the biased trajectory, or in plain words, making the training from a short biased trajectory equivalent to that of a long unbiased trajectory. We summarise hereafter the alternate approaches for processing the biased trajectories.

### No reweighting
Instead of reweighting, one can train an AE with the biased trajectory as is. This strategy is used in the MESA protocol (Chen *et al.,*

2018). With FEBILAE, it was shown, however, that ignoring the biases could introduce systematic errors (Belkacemi *et al.,* 2022).

### Direct reweighting
Assuming that the biasing potential along $\mathbf{s}$, $V_b(\mathbf{s})$, is suitably converged, then the unbiased time average of an observable $X_t$ of the process of interest can be computed as

$$\langle X_t \rangle = \frac{\int_0^t X_t e^{\beta V_b(\mathbf{s}_\tau)} \mathrm{d}\tau}{\int_0^t e^{\beta V_b(\mathbf{s}_\tau)} \mathrm{d}\tau}, \tag{8}$$

where $\beta = 1/k_\mathrm{B}T$, $k_\mathrm{B}$ is the Boltzmann constant and $T$ is the temperature of the simulation. Combining Eq. (8) with Eq. (1) yields the weighted MSE loss function, expressed as

$$\mathscr{L}_w = \frac{\sum_{j=0}^{N} w(j\Delta t) \|\mathbf{s}(j\Delta t) - \hat{\mathbf{s}}(j\Delta t)\|^2}{\sum_{j=0}^{N} w(j\Delta t)}, \tag{9}$$

where the weight for each time frame, $w(j\Delta t)$, is computed as $e^{\beta V_b(\mathbf{s}(j\Delta t))}$. Whereas this reweighting strategy is straightforward for classical AEs, and is utilised in FEBILAE (Belkacemi *et al.,* 2022), transposing it to TAEs, modified TAEs and VAMPnets is not obvious, since Eqs. (2), (3), (4) and (6) include correlations between two observable quantities that may carry different weights.

### Reweighting by uneven time intervals
Yang and Parrinello (2018) developed a numerical scheme by treating the biased trajectory as an unevenly spaced time series. Under these premises, the calculation of the elements in $\mathbf{C}(t, t+\tau)$ follows:

$$C_{ik}(t, t+\tau) = \frac{\sum_{j=0}^{N-\alpha} \Delta t_j \hat{\tilde{\xi}}_i(j\Delta t) \hat{\tilde{\xi}}_k(j\Delta t + \alpha\Delta t)}{\sum_{j=0}^{N-\alpha} \Delta t_j}, \tag{10}$$

and since the time intervals $\Delta t_j$ in an unbiased trajectory are even, the overlapping time between frames $j\Delta t$ and $j\Delta t + \alpha\Delta t$ is the same for all time-lagged pairs. As a result, the $\Delta t$ factors appearing in both the numerator and the denominator cancel out. In a biased trajectory, however, $\Delta t_j$ may differ between any two frames. To reweight the time series, we first need to compute the unbiased time interval for each frame, namely $\Delta t_j^* = \Delta t e^{\beta V_b(\mathbf{s}(j\Delta t))}$ (see also Eqs. (2)–(4) in Hamelberg *et al.,* 2004), and then find the overlapping time for each time-lagged pairs $\xi(j\Delta t)$ and $\xi(j\Delta t + \alpha\Delta t)$ by following the sequence of steps:

(1)  Compute the unbiased simulation time for each frame as a cumulative sum of $\Delta t_j^*$, namely, $t_j^* = \sum_{l=0}^{j} \Delta t e^{\beta V_b(\mathbf{s}(j\Delta t))}$;
(2)  Compute the unbiased lagged simulation time for each frame as $T_j^* = t_j^* + \tau$, where $\tau$ is the specified time lag;
(3)  Two monotonically increasing sequences $S_t = (t_0^*, \cdots, t_N^*)$ and $S_T = (T_0^*, \cdots, T_N^*)$ can be obtained from the previous steps. Now one can construct a union set of $S_t$ and $S_T$ with elements lying between $T_0^*$ and $t_N^*$, namely, $S_u = \left\{ t_j' \in S_t \cup S_T | t_j' > T_0^* \wedge t_j' < t_N^* \right\}$, and then sort $S_u$ by increasing order;

(4)   Assuming that $S_u$ has $M$ elements $t'_1, \cdots, t'_M$ in increasing order, for each element $t'_m$, we can find an index $i$ such that $t^*_i < t'_m < t^*_{i+1}$, and also an index $j$ such that $T^*_i < t'_m < T^*_{i+1}$. Since the finding of indices can be viewed as mappings or functions, then we obtain a list $L$ with $M-1$ elements including all triplets $\{f(m), g(m), \Delta t'_m\}$, where $f(m) = i$, $g(m) = j$ and $\Delta t'_m = t'_{m+1} - t'_m$ (see *Fig. 1c* for a schematized representation of the $\Delta t'_m$ calculation);

(5)   The reweighted correlation function can be calculated as:

$$C_{ik}(t, t+\tau) = \frac{\sum_{m=1}^{M-1} \Delta t'_m \hat{\tilde{\xi}}_i(f(m)) \hat{\tilde{\xi}}_k(g(m))}{\sum_{m=1}^{M-1} \Delta t'_m}. \quad (11)$$

### Koopman reweighting

Wu *et al.* (2017) proposed an approximation of the weighted correlation by means of Koopman reweighting, namely,

$$C_{ik}(t, t+\tau) \approx \frac{\sum_{j=0}^{N-\alpha} w(j\Delta t) \hat{\tilde{\xi}}_i(j\Delta t) \hat{\tilde{\xi}}_k(j\Delta t + \alpha \Delta t)}{\sum_{j=0}^{N-\alpha} w(j\Delta t)}. \quad (12)$$

In contrast with Eq. (11), Eq. (12) uses only the weight of time frame $j\Delta t$. This strategy has been used by McCarty and Parrinello (2017) for improving the selection of CVs with tICA. To determine a suitable reweighting strategy for the iterative learning, we compare the accuracy of Eqs. (11) and (12) for reweighting in numerical examples, as will be discussed in the following section.

### Results and discussion

In this section, we discuss through a series of numerical examples the limitation of AEs for learning slow modes, and how to introduce temporal information could address this limitation. Next, we review the possible shortcoming of using nonlinear activation functions in TAEs, and show how it can be dealt with, turning to either SRVs or an extension of modified TAEs. In addition, we assess different reweighting schemes consistent with SRVs and biased trajectories, thereby paving the way for iterative learning with CV-based enhanced sampling. Next, we illustrate our protocol in the discovery of CVs of two prototypical biological processes, namely the isomerization of NANMA and trialanine. Finally, we reveal the potential connection between the temporal models investigated herein with the committor probability (Geissler *et al.*, 1999) employed in transition path theory (Weinan and Vanden-Eijnden, 2010).

### AEs discover the high-variance modes instead of the slow modes

To illustrate the intrinsic limitations of classical AEs for capturing the most dominant features, we compared them and TAEs to learn a one-dimensional CV from overdamped Langevin dynamics trajectories with the following two-dimensional triple-well potential, using $X$ and $Y$ as the input features for NNs, setting $\alpha$ to 1.0 and to 10.0,

$$V(X, Y) = 3e^{-X^2}\left(e^{-\left(Y-\frac{1}{3}\right)^2/\alpha} - e^{-\left(Y-\frac{5}{3}\right)^2/\alpha}\right)$$
$$- 5e^{-Y^2/\alpha}\left(e^{-(X-1)^2} + e^{-(X+1)^2}\right) \quad (13)$$
$$+ 0.2X^4 + 0.2\left(Y - \frac{1}{3}\right)^4/\alpha^2.$$

The unit of variables $X$ and $Y$ is angstrom, and the unit of the output of $V(X, Y)$ is kcal mol$^{-1}$. The corresponding potential energy surfaces and the time evolution of the variables are shown in *Fig. 2a,b,e,f*. It can be easily deduced from *Fig. 2a,e* that for both $\alpha = 1.0$ and $\alpha = 10.0$, the important DOF should be $X$. As depicted in *Fig. 2c,g*, when training the trajectories by AEs, the value of the bottleneck layer, $\xi$, varies along $X$ if $\alpha$ is 1.0, but switches to vary along $Y$ if $\alpha$ increases to 10.0. This seemingly surprising result indicates that AEs erroneously learn two distinct important DOFs, when there should be only one along $X$. In stark contrast, TAEs behave consistently for both $\alpha = 1.0$ (*Fig. 2d*) and $\alpha = 10.0$ (*Fig. 2h*), and the learned variable highly correlates with the movement along

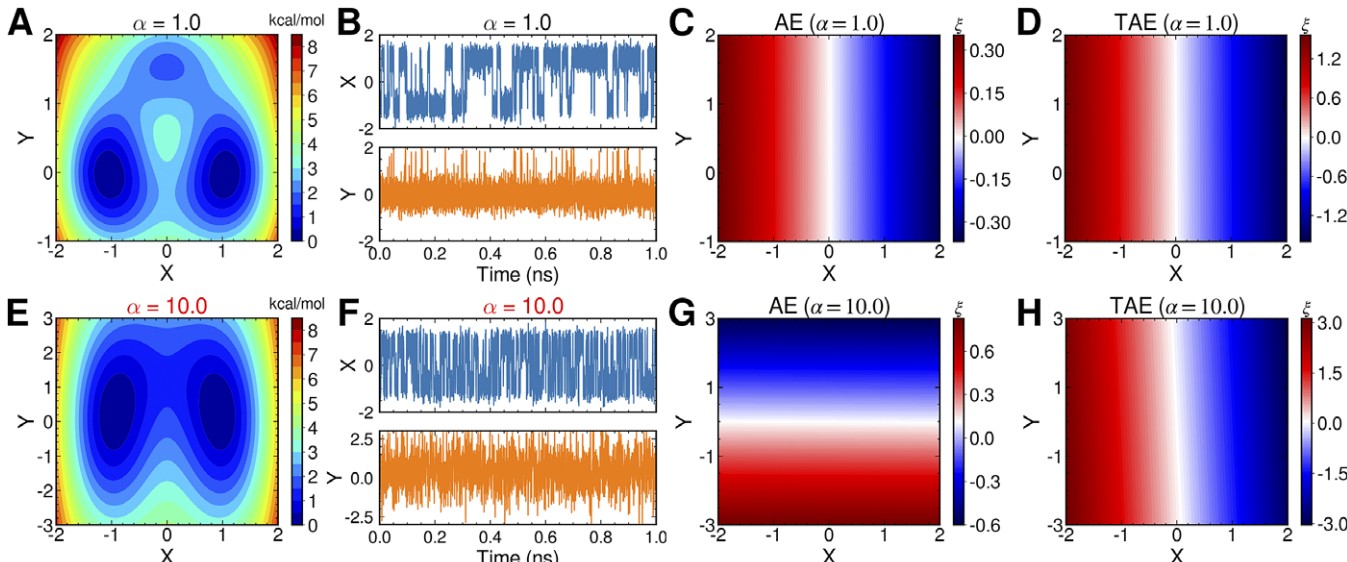

**Fig. 2.** Potential energy surfaces of $V(X, Y)$ with (a) $\alpha = 1.0$ and (e) $\alpha = 10.0$; Time evolution of $X$ and $Y$ when (b) $\alpha = 1.0$ and (f) $\alpha = 10.0$; Projections of the encoded variable $\xi$ on $X$ and $Y$ from AEs training with trajectories of (c) $\alpha = 1.0$ and (g) $\alpha = 10.0$; Projections of the encoded variable $\xi$ on $X$ and $Y$ from TAEs training with trajectories of (d) $\alpha = 1.0$ and (h) $\alpha = 10.0$.

$X$, which is also the most dominant mode. The projections of the one-dimensional learned variable onto $X$ and $Y$ of other models discussed in this contribution are shown in Supplementary Fig. 1. It can be rationalised by the fact that AEs actually learn the variables that correspond to the highest variance, which is, indeed, reflected in our analyses, with a variance for $X(t)$ of 1.0, greater than that for $Y(t)$, equal to 0.18, when $\alpha = 1.0$. Conversely, when $\alpha = 10.0$, the variance for $X(t)$ is 0.78, smaller than that of $Y(t)$, equal to 0.99. This discrepancy in the variances may also explain why AE fails to encode the correct important DOF when $\alpha = 10.0$. The present comparison underscores that it is crucial to introduce temporal information into the models for learning the slow modes, as opposed to the high-variance ones. Nevertheless, as discussed in a recent review of Markov state models, sometimes slow modes may still not coincide with the important DOFs that are sought (Husic and Pande, 2018). We, therefore, need to select proper candidate CVs, or features, as the learning input, and that is why we advocate here to combine data-driven and intuition-based approaches.

### Both nonlinear modified TAE and SRVs can correctly learn the slow modes

Chen *et al.* (2019*a*) have illustrated that nonlinear TAEs learn a mixture of slow modes and high-variance modes, and proposed the nonlinear modified TAEs that can learn correctly the slow modes, but comparing to SRVs, modified TAEs cannot generate orthogonal and multidimensional latent variables that are suitable for CV-based biased simulations. Here, we confirm their results numerically by using the potential depicted in *Fig. 2c*, but we also propose that the modified TAE can be extended to slow feature analysis (SFA) (Wiskott and Sejnowski, 2002; Berkes and Wiskott, 2005; Song and Zhao, 2022) (see *Fig. 1b*), which can render the latent variables orthogonal in multidimensional cases, since Eq. (4) resembles the SFA loss function in the one-dimensional case. In an SFA, if the encoder, $\mathbf{f}$, is multivariate, Eq. (4) should be formulated as the following generalised eigenvalue problem:

$$\begin{cases} \mathbf{A} &= \dfrac{1}{N-\alpha}[\hat{\mathbf{X}}(t+\tau) - \hat{\mathbf{X}}(t)][\hat{\mathbf{X}}(t+\tau) - \hat{\mathbf{X}}(t)]^{\mathsf{T}} \\ \mathbf{AV} &= \mathbf{C}(t,t)\mathbf{V}\mathbf{\Lambda}. \end{cases} \quad (14)$$

The corresponding loss function is the sum of squared eigenvalues in Eq. (14), namely, $\mathscr{L} = \sum_{i=1}^{m} \lambda_i^2$. Akin to SRVs, the CVs are then expressed as linear combinations of the components of $\mathbf{f}$. We note that our use of SFA in conjunction with NNs is quite similar to the deep SFA (DSFA) for change detection in images (Du *et al.*, 2019), except that in our two encoders, the weights and biases are shared. At first glance, Eq. (14) looks similar to Eq. (7), and also the loss function of SFA resembles that of SRVs, except that the former does not have a negative sign. Indeed, in the linear case, the two methods have been proved to be equivalent (Blaschke *et al.*, 2006; Wang and Zhao, 2020), but the relationship is not so clear in the nonlinear case, whereby a NN with nonlinear activation functions is employed to encode $\hat{\mathbf{X}}$. Consequently, to investigate the performance of finding nonlinear latent variables for slow modes, in this section, we benchmark a TAE, modified TAE, SRVs and SFA using the unbiased trajectory generated from the potential energy function Eq. (13), with $\alpha = 10.0$. We used two hidden layers with hyperbolic tangent as the activation functions in both the encoder and the decoder part of TAEs, giving a final architecture of 2-40-40-$n$-40-40-2. The activation functions of the input, the bottleneck and the output layer are linear. Similarly, we used an

architecture of 2-40-40-$n$ for the other Siamese NNs (modified TAE, SFA and SRVs). We have tested both cases, where $n = 1$ and $n = 2$, and used a time lag of 0.02 ps.

The projections of the encoded values in the one-dimensional case – that is, the outputs of the bottleneck layer in the TAE, the last layers in the modified TAE and the linear combinations of basis functions of SFA and SRVs – are gathered in *Fig. 3a–d*. In *Fig. 3a*, we can clearly see that the latent variable of the TAE is split into the left and right sides along $X$, and on each side, $\xi$ varies along $Y$. This result implies that while the nonlinear TAE is able to learn a latent variable, distinguishing the two basins shown in *Fig. 2c*, it is also affected by the high-variance mode in $Y$. In stark contrast, the modified TAE, SFA and SRVs, with nonlinear activation functions, appropriately learn the slow modes, that is, the learned CV varies almost only along the transition between the two basins. In the two-dimensional case, comparing *Fig. 3f,j*, we note that the modified TAE learns two nearly identical modes, both varying along $X$. The generalisation of the modified TAE, using the SFA loss function embodied in Eq. (14), is, however, able to recognise two distinct, approximately orthogonal modes (see *Fig. 3g,k*), in line with the results of the SRVs (see *Fig. 3h,l*).

Furthermore, we compared the two reweighting schemes, namely reweighting by uneven time intervals (Eq. (11)) and Koopman reweighting (Eq. (12)), for the calculation of the weighted correlation in SRVs from an egABF biased trajectory, with $\alpha = 1.0$. The variations of the two learned CVs, $\xi_1$ (*Fig. 3q*) and $\xi_2$ (*Fig. 3r*), using Koopman reweighting, closely correlate with the reference ones (*Fig. 3m,n*) obtained from an unbiased trajectory. The results from the reweighting strategy with uneven time intervals (see *Fig. 3o,p*) clearly depart from the reference ones. Hence, we chose the Koopman reweighting scheme for learning CVs from biased simulations in our iterative approach. The accuracy of the scheme leaning on uneven time intervals for the evaluation of the correlations may, nevertheless, be improved by means of interpolations, either explicitly or implicitly via Fourier transform (Scargle, 1989), which falls beyond the scope of the present contribution.

### Iterative learning of the CVs for the isomerizations of NANMA and trialanine

From the benchmarks of the previous two sections, we have concluded that SRVs are able to find both slow and orthogonal modes in multidimensional cases, and SFA performs similarly. In this section, we further test SRVs (Chen *et al.*, 2019*b*) for the purpose of CV discovery based on biased simulations, applied specifically to the isomerization of NANMA and of trialanine, both in vacuum. Both peptides have been widely used as case examples in the development of novel enhanced-sampling and path-searching methods (Pan *et al.*, 2008; Hénin *et al.*, 2010; Branduardi *et al.*, 2012; Valsson and Parrinello, 2014; Tiwary and Berne, 2016; Chen *et al.*, 2022*a*). In contrast with a previous study that uses deep-TICA in a single iteration (Bonati *et al.*, 2021) to find from a trajectory with biasing potentials the slow mode along $\phi$ only, we employed an iterative learning approach, akin to FEBILAE (Belkacemi *et al.*, 2022), using an initial trajectory from an egABF (Zhao *et al.*, 2017) biased simulation, and performed Koopman reweighting (see Eq. (12); Wu *et al.*, 2017), as described in the Methods section. The guidelines for choosing the NN hyperparameters, the parameters of the iterative learning and the simulation details can be found in the Supplementary Material.

From the reference free-energy landscape of NANMA along backbone angles $\phi$ and $\psi$ (*Fig. 4a*), we are able to identify the

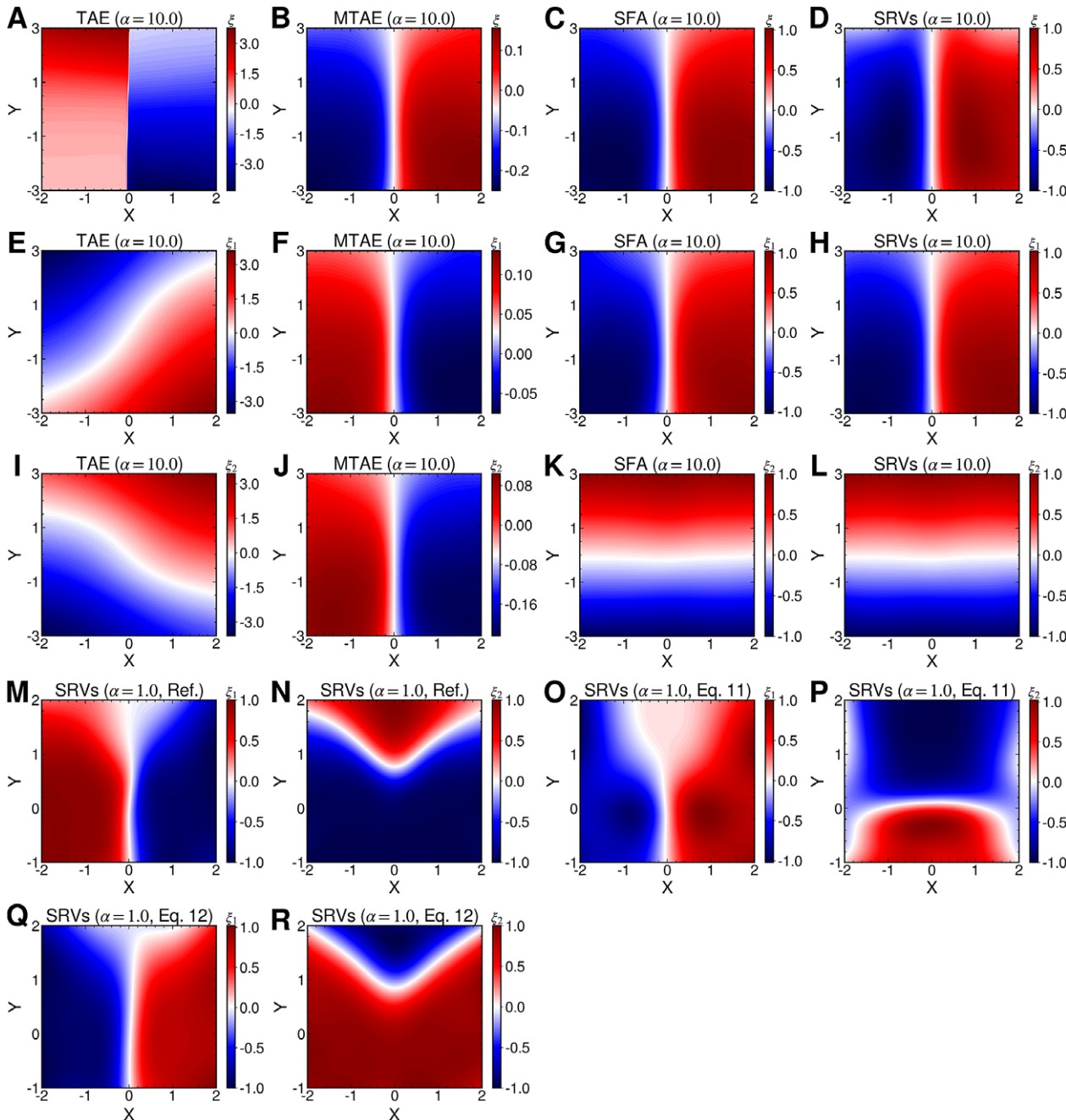

**Fig. 3.** Projections of the one-dimensional CV $\xi$ learned from unbiased trajectories when $\alpha = 10.0$ on $X$ and $Y$ of (a) TAE, (b) modified TAE, (c) SFA and (d) SRVs. Projections of the 2D CVs $(\xi_1, \xi_2)$ learned from unbiased trajectories when $\alpha = 10.0$ on $X$ and $Y$ of (e,i) TAE, (f,j) modified TAE, (g,k) SFA and (h,l) SRVs. Projections of the 2D CVs $(\xi_1, \xi_2)$ learned by SRVs from an unbiased trajectory when $\alpha = 10.0$ (m,n), from an egABF biased trajectory reweighted by Eq. (11) (o,p), and from an egABF biased trajectory reweighted by Eq. (12) (q,r).

minimum free-energy pathway connecting states $C_{7ax}$ and $C_{7eq}$, via the extended form, $C_5$ (shown as grey dots in *Fig. 4b*), turning to the multidimensional lowest energy (MULE) algorithm (Fu *et al.,* 2020). The projection of the learned CV $\xi_1$ onto $\phi$ and $\psi$ in *Fig. 4d* clearly distinguishes the $C_{7ax}$ and $C_{7eq}$ states, and interprets $C_5$ as an intermediate state. We can also identify three basins on the PMF along the learned CV $\xi_1$ (*Fig. 4c*), and by analysing the MD trajectories we have found these basins corresponding, indeed, to $C_{7eq}$, $C_5$ and $C_{7ax}$, respectively. Moreover, the free-energy difference between $C_{7eq}$ and $C_{7ax}$ obtained from *Fig. 4c* is equal to 2.1 kcal·mol$^{-1}$, which only deviates slightly from the reference result (2.3 kcal·mol$^{-1}$) deduced from *Fig. 4b*. The free-energy difference between $C_{7eq}$ and

$C_5$ obtained from *Fig. 4c* amounts to 1.2 kcal·mol$^{-1}$, which is also very close to the reference result (1.0 kcal·mol$^{-1}$). Additionally, the major free-energy barrier, separating $C_5$ from $C_{7ax}$ in *Fig. 4c*, on the PMF along $\xi_1$ is equal to 8.2·kcal mol$^{-1}$, which marginally deviates from the ground-truth reference (8.1 kcal·mol$^{-1}$, marked by the red circle in *Fig. 4b*). In summary, not only is the learned CV, $\xi_1$, able to discriminate qualitatively between the different metastable states encountered in the isomerization of NANMA, but the PMF along this learned variable also quantitatively predicts the correct free-energy difference and barrier.

In the paradigmatic case of NANMA, the selected candidate CVs actually coincide with the physically correct ones, namely $\phi$

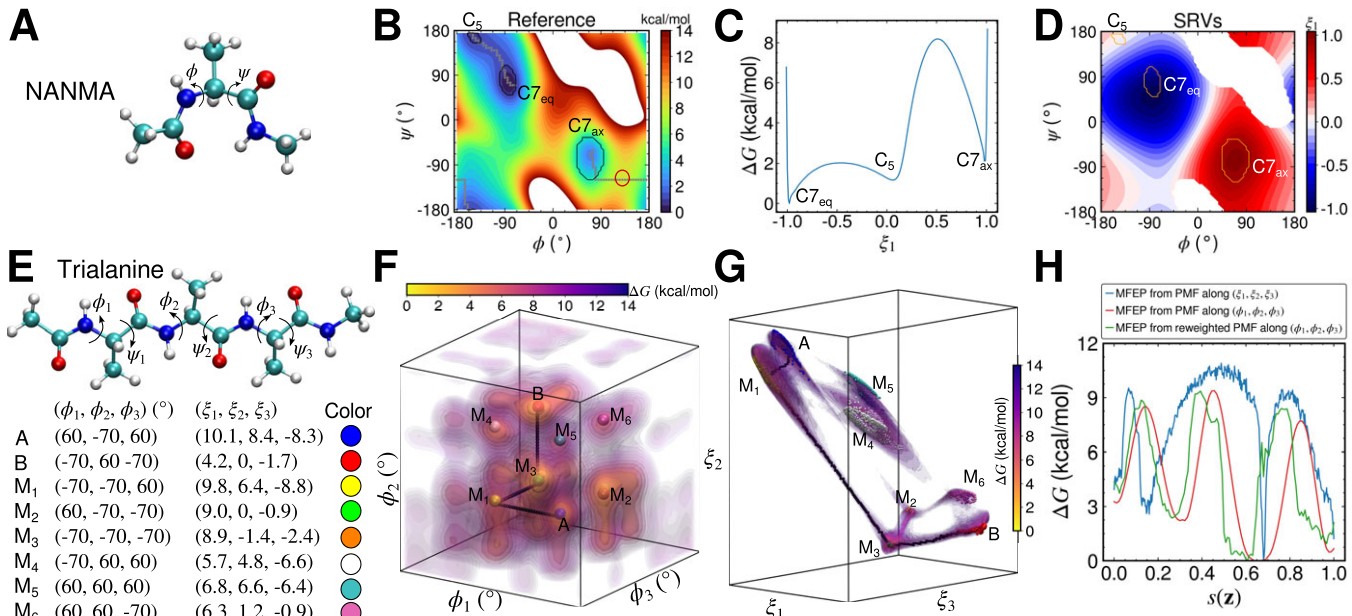

**Fig. 4.** (*a*) Structure of NANMA and two dihedral angles $\phi$ and $\psi$ as candidate CVs. (*b*) Reference free-energy landscape of the NANMA isomerization along $\phi$ and $\psi$. The grey dots show the minimum free-energy pathway (MFEP) from $C_{7eq}$ to $C_{7ax}$ via $C_5$. The dominant free-energy barrier on the MFEP is marked by the red circle. (*c*) The PMF along the learned CV $\psi_1$. The three basins correspond to $C_{7eq}$, $C_5$ and $C_{7ax}$, respectively. (*d*) The value of learned CV $\xi_1$ projected on $\phi$ and $\psi$. (*e*) Structure of trialanine, the candidate CVs ($\phi_1, \psi_1, \phi_2, \psi_2, \phi_3, \psi_3$), and the eight basins (A, B, $M_1$, $M_2$, $M_3$, $M_4$, $M_5$, $M_6$) found in the free-energy landscapes along the reference CVs in (*f*) and the learned CVs in (*g*). The basins A, B, $M_1$, $M_2$, $M_3$, $M_4$, $M_5$ and $M_6$ are marked in blue, red, yellow, green, orange, white, cyan and pink, respectively. (*f*) Reference 3D free-energy landscape along ($\phi_1, \phi_2, \phi_3$) and the corresponding MFEP (black dots). (*g*) 3D free-energy landscape along the learned CVs ($\xi_1, \xi_2, \xi_3$) and the corresponding MFEP (black dots). (*h*) MFEPs found from the reference free-energy landscape along ($\phi_1, \phi_2, \phi_3$) (red), the free-energy landscape along the learned CVs ($\xi_1, \xi_2, \xi_3$) (blue), and the free-energy landscape reweighted from ($\xi_1, \xi_2, \xi_3$) to ($\phi_1, \phi_2, \phi_3$) (green).

and $\psi$. Had we only a limited knowledge of the underlying dynamics of the process at hand, and had we included some candidate CVs that are not relevant, could our protocol still be able to learn the correct CVs? To answer this question, we tackled the more challenging example of trialanine, for which we pretend that we do not know that the three $\phi$ dihedral angles are important (Valsson and Parrinello, 2014; Tiwary and Berne, 2016) to its isomerization, and blindly select all the backbone, $\phi$ and $\psi$, angles (see *Fig. 4a*) to form the candidate CVs, and see whether the learned CVs can render a satisfactory picture of the conformational changes. The ground-truth reference three-dimensional free-energy landscape along the three known important CVs ($\phi_1, \phi_2, \phi_3$) is shown in *Fig. 4f*, and eight metastable states can be identified in the basins marked as A, B, $M_1$, $M_2$, $M_3$, $M_4$, $M_5$ and $M_6$. Fig. 4 depicts the three-dimensional free-energy landscape along the learned CVs ($\xi_1, \xi_2, \xi_3$), where we can also identify eight basins. After analysis of the trajectory, we have discovered that these basins correspond to the conformations A, B and $M_1-M_6$, which indicates that the learned CVs are able to discriminate between the important conformations. Moreover, by applying MULE on the three-dimensional free-energy landscape along ($\xi_1, \xi_2, \xi_3$), we determined the minimum free-energy pathway as A-$M_1$-$M_3$-B (shown as black spheres in *Fig. 4g*), which coincides with that found in the reference three-dimensional free-energy landscape along ($\phi_1, \phi_2, \phi_3$) (shown as black spheres in *Fig. 4f*). It ought to be noted that a previous study (Chen *et al.*, 2022b) demonstrated that pathway A-$M_1$-$M_3$-B also corresponds to the most probable transition pathway (MPTP) (Pan *et al.*, 2008). The free energies determined along the MFEP from the three-dimensional free-energy calculation along ($\xi_1, \xi_2, \xi_3$) and the reference is shown in *Fig. 4h* in blue and red, respectively. The deviation between the blue and red curves may stem from discretization issues and difficulty to enhance sampling in the three-dimensional space (see Supplementary Material for details).

However, if we reweight the free-energy landscape along ($\xi_1, \xi_2, \xi_3$) to that along ($\phi_1, \phi_2, \phi_3$), and identify the MFEP again, we can observe that the resulting profile (green curve in *Fig. 4h*) is very close to the reference one (red curve in *Fig. 4h*). We, therefore, conclude that the CVs ($\xi_1, \xi_2, \xi_3$) obtained from SRVs with iterative learning reproduce the correct dynamics underlying the isomerization of trialanine, even if some fast and non-important candidate CVs are included.

Comparing the NN hyperparameters used in the NANMA and the trialanine test cases as shown in Supplementary Table 1, we can see that more neurons or computational units are used in the latter case. Therefore, if the presented approach is applied to larger biological objects, it is likely that the choice of candidate CVs differs from those for NANMA and trialanine, and the NNs will become more complex through (a) an increase of the number of layers and neurons, and (b) combination of different types of layers, for example, dropout and convolution layers, for training.

### Potential connections between TAE, modified TAE and committor

In transition path theory (Weinan and Vanden-Eijnden, 2010), considering that a molecular process characterised by two metastable states *A* and *B*, the net forward reactive flux from state *A* to state *B* can be expressed as a time-correlation function (Krivov, 2021; Roux, 2021),

$$J_{AB} = \frac{1}{2\tau}\langle[q(\mathbf{s}(t+\tau)) - q(\mathbf{s}(t))]^2\rangle, \qquad (15)$$

where the committor, $q(\mathbf{s})$, is the sum of the probability over all paths initiating from **s** that ultimately reach *B* before visiting *A*. By definition, $q(\mathbf{s}_A)$ and $q(\mathbf{s}_B)$ are equal to 1 and 0, respectively, and as a sum of probability, $q$ should be bounded, namely $q \in [0,1]$.

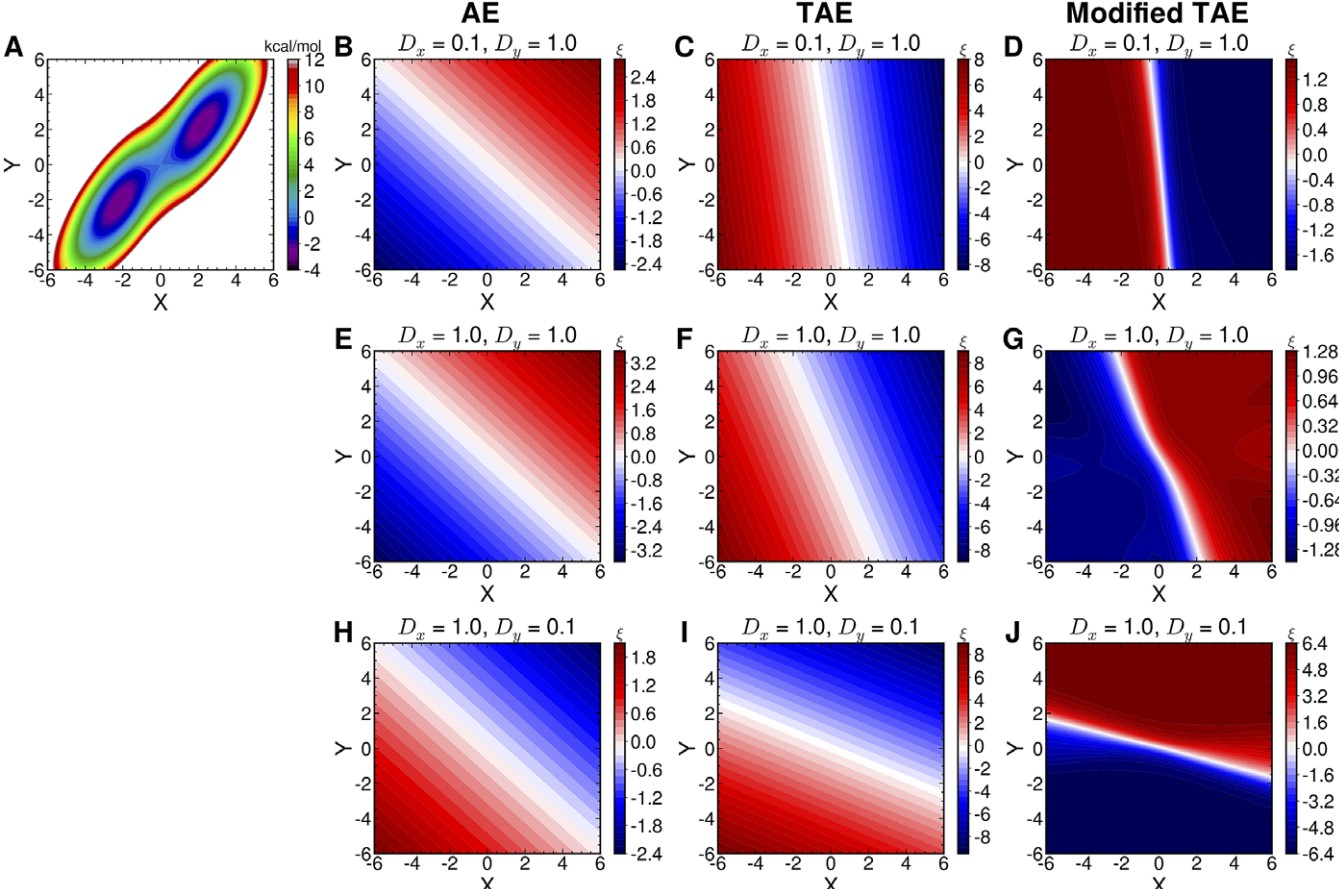

**Fig. 5.** (*a*) Berezhkovskii–Szabo potential energy surface (Berezhkovskii and Szabo, 2005). The latent variables projected onto (*X, Y*) learned by AEs, TAEs and modified TAEs in three diffusivity conditions: (*b–d*) $D_x/D_y = 0.1$, (*e–g*) $D_x/D_y = 1.0$ and (*h–j*) $D_x/D_y = 10.0$. The AEs and TAEs are trained with a neural network architecture of 2-10-1-10-2 with linear activation functions used in all layers. The modified TAEs are trained with a 2-20-20-1 neural network, and the hyperbolic tangent is used as the activation functions for the two hidden layers with 20 computational units. The time lag for TAEs and modified TAEs is 10 steps.

Similar to the minimization in Eq. (4) for learning CVs, it can be envisioned that the committor function $q(\mathbf{s})$ can also be obtained by the minimization of Eq. (15) with the following restraints,

$$q(\mathbf{s}(t)) = \begin{cases} 0.0, & \mathbf{s}(t) \in A \\ f_\theta(\mathbf{s}(t)), & \mathbf{s}(t) \notin (A \cup B), \\ 1.0, & \mathbf{s}(t) \in B \end{cases} \quad (16)$$

where $f_\theta(\mathbf{s}(t))$ is the output of a one-dimensional NN-based function with $\theta$ as its parameters. Based on such a minimization, He *et al.* (2022) have recently proposed the committor-consistent variational string (CCVS) method, where $f_\theta(\mathbf{s}(t))$ is parametrized by a linear combination of basis functions constructed from Voronoi cells supported by images of strings, and then optimised by an iterative Monte-Carlo procedure, as a way to determine the underlying transition pathway. At first glance, Eq. (15) looks similar to the numerator in Eq. (4) multiplied by a constant in one-dimensional cases. The main difference is that Eq. (15) is not scaled by the variance of $f_\theta(\mathbf{s}(t))$ or $q(\mathbf{s}(t))$. Instead, as shown in Eq. (16), the committor-based loss function explicitly requires boundary conditions to identify the two metastable states, A and B, before training. In stark contrast, the loss function of modified TAEs in Eq. (4) does not feature boundary conditions, and works as a method for blind separation. The CCVS authors have further demonstrated that their method is sensitive to the diffusivities of

the components in **s**, for example, the anisotropic diffusivities along $X$ and $Y$ of the Berezhkovskii–Szabo potential (*Fig. 5a*) resulting in different isocommittors (Weinan *et al.*, 2005).

As we can see, Eq. (15) resembles the loss function of modified TAEs embodied in Eq. (4) if $\hat{\xi}$ is one-dimensional. We found the similar manifestation of the anisotropic diffusivity in the CCVS (He *et al.*, 2022) and in the learned CV intriguing, as it suggests the possibility of encoding kinetics information, such as diffusivities, in the methodology discussed in the present contributions. Here, we present a preliminary investigation of this hypothesis, wherein AEs, TAEs, and modified TAEs are compared, using the Brownian dynamics trajectories of anisotropic diffusivities sampled from the Berezhkovskii–Szabo potential (Berezhkovskii and Szabo, 2005) as the training inputs. With the diffusivities along $X$ and $Y$ denoted $D_x$ and $D_y$, respectively, we have examined three cases, namely $D_x/D_y = 0.1$, $D_x/D_y = 1.0$ and $D_x/D_y = 10.0$. The results of the encoded variables $\xi$ projected back onto the $(X, Y)$ plane are gathered in *Fig. 5b–j*. Comparing *Fig. 5b,e,h*, we find that the results from the classical AE are invariant to the change of $D_x/D_y$. In stark contrast, in the cases of the TAEs (see *Fig. 5c,f,i*) and modified TAEs (see *Fig. 5d,g,j*), the learned CVs are affected by the different $D_x/D_y$ ratios, which indicates that the time-series-based models are capable of reflecting the anisotropic diffusivity, and may have potential connections with the CCVS in the case of a two-state molecular process. Additionally, since AEs, TAEs and modified TAEs do not

utilise restraints similar to those in Eq. (16), the learned CVs at a specific metastable state do not have a fixed value. In other words, for a committor $q$, we have $q(\mathbf{s}_A) = 0$ and $q(\mathbf{s}_B) = 1$, but for a good learned CV $\xi$, we only know that $\xi(\mathbf{s}_A) \neq \xi(\mathbf{s}_B)$. The exact values of $\xi(\mathbf{s}_A)$ and $\xi(\mathbf{s}_B)$ are affected by the randomisation of the initial parameters and optimizers, which explains the colour flipping in *Fig. 5h*, contrasting with *Fig. 5b,e*.

## Conclusion

In this contribution, we have reviewed the counterparts of PCA and TICA in the era of deep learning, including AEs, TAEs, modified TAEs, SFA and SRVs, examined the limitations of classical AEs through a series of numerical examples, and confirmed that classical AEs capture the high-variance modes in lieu of the slow modes. This limitation can, however, be overcome by turning to time-series-based models, such as TAEs, modified TAEs and SRVs. Our numerical experiments, nevertheless, confirm that in nonlinear cases, TAEs still encode a mixture of high-variance and slow modes, which can be circumvented by turning to modified TAEs and SRVs. Given that the original form of the modified TAEs cannot adequately learn the orthogonal latent variables, we have proposed an extension of the latter models that expand modified TAEs by SFA, able to yield orthogonal latent variables. In order to combine time-series-based models with iterative learning and enhanced-sampling based free-energy calculations, we have examined and compared critically alternate reweighting schemes that enable models to be trained from biased trajectories, while preserving the underlying unbiased slow modes of the molecular process at hand, thereby paving the way for iterative learning with enhanced-sampling algorithms. As an illustration, we have employed our proposed iterative-learning protocol to discover the CVs describing the isomerization of both NANMA and trialanine, and showed that the free-energy landscapes along the learned CVs feature the correct metastable states, allowing the minimum free-energy pathways to be identified, and the free-energy barriers to be computed. In addition, we have probed time-series-based models in the case of anisotropic diffusivity and found that the learned CV may have deep connections with CCVS, thus suggesting that these models might be eminently relevant to learn committor probabilities. To summarise, armed with a proper reweighting method, SRVs with iterative learning are well-suited for the discovery of CVs, which can be subsequently utilised in free-energy calculations. It ought to be noted, however, that appropriateness of the time-series-based models is still subservient to a reasonable choice of the hyperparameters that control the NN, like the time lag, the number of hidden layers, and the number of computational units – or neurons – in each layer, which may affect the resolution of slowness and the degree of overfitting. Possible solutions to address this issue may include integrating multiple time lags (Lorpaiboon *et al.,* 2020; Wang and Zhao, 2020), or using singular spectrum analysis (Hassani, 2007), which inherently employs multiple time lags. Furthermore, in order to avoid overfitting and render the models more robust with respect to unseen or missing data, one may incorporate into the SRVs probabilistic models, like a probabilistic SFA (Turner and Sahani, 2007) and a predictive information bottleneck, or PIB (Wang *et al.,* 2019), with the help of invertible neural networks, or INNs (Ardizzone *et al.,* 2019).

**Open peer review.** To view the open peer review materials for this article, please visit http://doi.org/10.1017/qrd.2022.23.

**Supplementary materials.** To view supplementary material for this article, please visit http://doi.org/10.1017/qrd.2022.23.

**Data availability statement.** The data that support the findings of this study are openly available upon request.

**Acknowledgements.** The authors are grateful to Tony Lelièvre and Gabriel Stoltz (École des Ponts ParisTech), Paraskevi Gkeka (Sanofi), Andrew Ferguson and Benoit Roux (University of Chicago) and Luigi Bonati (Italian Institute of Technology) for insightful discussions.

**Author contributions.** H.C. and C.C. designed the study, performed the simulations, analysed the data and wrote the manuscript.

**Funding statement.** This work was supported by the Agence Nationale de la Recherche (Lorraine Artificicial Intelligence – LOR-AI and ProteaseInAction).

**Conflict of interest.** The authors declare no conflicts of interest.

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
