## [Reviewer Report]

*Comments to Author*: The manuscript by Chen and Chipot compares existing a number of machine-learning methods that aim at discovering collective variables for free energy calculations. Specifically, this work focuses on auto-encoders (AEs) as well as time-series-based variants, including time-lagged AEs (TAEs) and modified TAEs. The manuscript is well-written, although too technical, and provides a short and clear description of the methods used to compare these methods in a few toy models. I find the manuscript of interest to the community due to the fair comparison it provides between AE, TAE, and modified TAE. However, there are still some points that need to be addressed:

1) This work only deals with very simple toy models. I think the authors can at least discuss or speculate on the performance of these methods when larger systems such as proteins and large-scale conformational changes are studied using such methods? For instance, is there a difference in the complexity of these algorithms as the number of DOFs increases?

2) The last part of the Results section (Potential connections between TAE, modified TAE, and committor) is somewhat too short and unclear. It is an interesting section but the authors need to make the connection to the rest of manuscript more clear and potentially expand on it. The committor function is suggested as “the reaction coordinate” in the transition path theory literature. The modified TAE (like the other algorithms discussed) tried to identify the most relevant collective variable as well. There is also similarity between Relations (4) and (15). However, the manuscript does not attempt to go beyond noticing the similarity and directly jumps to a numerical example. I think some theoretical work or at least deep discussion is missing here before this jump.

More minor points:

1) Fig. 5: Why the color is flipped in H as opposed to B and E?

2) Page 2: “the variables that can maximize the explained variances do not always necessarily coincide with the DOFs of the process of interest” (I feel adding “important” before DOFs or replacing “of” with “relevant to” makes this clearer)

3) Page 2: “observationraises” (typo)

4) Eq.1: Is ∆t timestep? Is this the same timestep used in MD simulations. If not, please use a different term.

5) Page 6: the line after Eq. 5 states “C00, C01 and C11 are defined …” (Are these supposed to be “C(t,t), C(t,t+τ), and C(t+τ,t+τ)“?)

6) Page 10, Relation (13): There is some inconsistencies with the units. Since potential is shown in kcal/mol, this should somehow come out of Eq. 13 but it does not.

7) Page 10: “start contrast” must be “stark contrast”

---

## [Reviewer Report]

*Comments to Author*: In this work, the authors review and examine the deep learning based methods in collective variable (CV) discovery, including AE, TAE, modified TAE, SRV, and SFA. Experiments unveil that AE learns high-variance modes instead of slow modes and TAE learns the mixture of these two modes. Modified TAE, SFA, and SRV appropriately learn the slow modes. Further experiments on NANMA and trialanine show the reweighting schemes enable deep learning models to learn CVs from biased trajectories. Overall, this work is well motivated and easy to follow. It also includes convincing experiments to evaluate different deep learning based methods for CVs. I only have following a few mild comments before the paper get published.

1. How are the architectures of the deep neural networks determined. For instance, NANMA uses 4-12-10-8-6-4-2, which are quite a few layers considering the small number of neurons per layer. Also, why are tanh used as activation instead of more widely used ReLU-ish functions?

2. In page 15, the authors mention “The deviation between the blue and red curves may stem from discretization issues and difficulty to enhance sampling in the three-dimensional space.” Could the authors elaborate more on what the discretization issues are.

3. The authors include experiments on a triple-well potential, NANMA, and trialanine which are convincing. However, these are still low-dimensional problems compared with molecular simulations in practice. How do the authors comment on the generalization of the reviewed methods on larger systems?

4. How will deep learning models (e.g., TAE, modified TAE, etc.) perform if the dimension of latent space is smaller than the actual CVs? Can the models learn the most dominant variables automatically?

---

## [Reviewer Report]

*Comments to Author*: Reviewer #2: In this work, the authors review and examine the deep learning based methods in collective variable (CV) discovery, including AE, TAE, modified TAE, SRV, and SFA. Experiments unveil that AE learns high-variance modes instead of slow modes and TAE learns the mixture of these two modes. Modified TAE, SFA, and SRV appropriately learn the slow modes. Further experiments on NANMA and trialanine show the reweighting schemes enable deep learning models to learn CVs from biased trajectories. Overall, this work is well motivated and easy to follow. It also includes convincing experiments to evaluate different deep learning based methods for CVs. I only have following a few mild comments before the paper get published.

1. How are the architectures of the deep neural networks determined. For instance, NANMA uses 4-12-10-8-6-4-2, which are quite a few layers considering the small number of neurons per layer. Also, why are tanh used as activation instead of more widely used ReLU-ish functions?

2. In page 15, the authors mention “The deviation between the blue and red curves may stem from discretization issues and difficulty to enhance sampling in the three-dimensional space.” Could the authors elaborate more on what the discretization issues are.

3. The authors include experiments on a triple-well potential, NANMA, and trialanine which are convincing. However, these are still low-dimensional problems compared with molecular simulations in practice. How do the authors comment on the generalization of the reviewed methods on larger systems?

4. How will deep learning models (e.g., TAE, modified TAE, etc.) perform if the dimension of latent space is smaller than the actual CVs? Can the models learn the most dominant variables automatically?

Reviewer #3: The manuscript by Chen and Chipot compares existing a number of machine-learning methods that aim at discovering collective variables for free energy calculations. Specifically, this work focuses on auto-encoders (AEs) as well as time-series-based variants, including time-lagged AEs (TAEs) and modified TAEs. The manuscript is well-written, although too technical, and provides a short and clear description of the methods used to compare these methods in a few toy models. I find the manuscript of interest to the community due to the fair comparison it provides between AE, TAE, and modified TAE. However, there are still some points that need to be addressed:

1) This work only deals with very simple toy models. I think the authors can at least discuss or speculate on the performance of these methods when larger systems such as proteins and large-scale conformational changes are studied using such methods? For instance, is there a difference in the complexity of these algorithms as the number of DOFs increases?

2) The last part of the Results section (Potential connections between TAE, modified TAE, and committor) is somewhat too short and unclear. It is an interesting section but the authors need to make the connection to the rest of manuscript more clear and potentially expand on it. The committor function is suggested as “the reaction coordinate” in the transition path theory literature. The modified TAE (like the other algorithms discussed) tried to identify the most relevant collective variable as well. There is also similarity between Relations (4) and (15). However, the manuscript does not attempt to go beyond noticing the similarity and directly jumps to a numerical example. I think some theoretical work or at least deep discussion is missing here before this jump.

More minor points:

1) Fig. 5: Why the color is flipped in H as opposed to B and E?

2) Page 2: “the variables that can maximize the explained variances do not always necessarily coincide with the DOFs of the process of interest” (I feel adding “important” before DOFs or replacing “of” with “relevant to” makes this clearer)

3) Page 2: “observationraises” (typo)

4) Eq.1: Is ∆t timestep? Is this the same timestep used in MD simulations. If not, please use a different term.

5) Page 6: the line after Eq. 5 states “C00, C01 and C11 are defined …” (Are these supposed to be “C(t,t), C(t,t+τ), and C(t+τ,t+τ)”?)

6) Page 10, Relation (13): There is some inconsistencies with the units. Since potential is shown in kcal/mol, this should somehow come out of Eq. 13 but it does not.

7) Page 10: “start contrast” must be “stark contrast”